# Thermally Mendable Self-Healing Epoxy Coating for Corrosion Protection in Marine Environments

**DOI:** 10.3390/ma16051775

**Published:** 2023-02-21

**Authors:** Eugenio Amendola, Barbara Palmieri, Stefania Dello Iacono, Alfonso Martone

**Affiliations:** 1Institute of Polymers, Composites and Biomaterials (IPCB), National Research Council, P.le Enrico Fermi 1, 80055 Portici, Italy; 2Institute of Applied Sciences and Intelligent Systems (ISASI), National Research Council, Via P. Castellino 111, 80131 Napoli, Italy

**Keywords:** Diels–Alder, self-healing, corrosion, nanoindentation

## Abstract

Polymeric coatings represent a well-established protection system that provides a barrier between a metallic substrate and the environment. The development of a smart organic coating for the protection of metallic structures in marine and offshore applications is a challenge. In the present study, we investigated the use of self-healing epoxy as an organic coating suitable for metallic substrates. The self-healing epoxy was obtained by mixing Diels–Alder (D–A) adducts with a commercial diglycidyl ether of bisphenol-A (DGEBA) monomer. The resin recovery feature was assessed through morphological observation, spectroscopic analysis, and mechanical and nanoindentation tests. Barrier properties and anti-corrosion performance were evaluated through electrochemical impedance spectroscopy (EIS). The film on a metallic substrate was scratched and subsequently repaired using proper thermal treatment. The morphological and structural analysis confirmed that the coating restored its pristine properties. In the EIS analysis, the repaired coating exhibited diffusive properties similar to the pristine material, with a diffusivity coefficient of 1.6 × 10^−6^ cm^2^/s (undamaged system 3.1 × 10^−6^ cm^2^/s), confirming the restoration of the polymeric structure. These results reveal that a good morphological and mechanical recovery was achieved, suggesting very promising applications in the field of corrosion-resistant protective coatings and adhesives.

## 1. Introduction

Protective coatings are widely utilized to promote corrosion resistance in the surfaces of steel components used in various industrial fields. Therefore, the development of systems capable of preventing corrosion damage to metallic substrates is a fundamental problem across many fields of application. Commonly, polymeric composites are used as continuous phases to form a barrier against moisture and corrosive solutions. Inevitably, such protective layers tend to fail over time, due to aging and environmental/mechanical damage. Although not visible to the naked eye, nano, and micro defects create discontinuity in the polymer film. If not repaired promptly, micro-damage evolves into a severe reduction in the effectiveness of the corrosion resistance. Recently, to avoid the costly replacement of damaged coated metal objects, substantial efforts have been focused on the development of smart polymer materials with self-healing properties, which can significantly prolong their service life [1,2]. The polymeric resin acts as a barrier against the corrosion of metallic substrates: epoxies and polyurethanes binders ensure good adhesion to metals, and the prevention of degradation occurs if the integrity of the coating is preserved. Several parameters contribute to the performance of the binder, such as the polymeric structure (backbone and side chains), the degree of crosslinking, and molecular relaxations (physical aging) [3].

A first attempt to produce smart protective coatings was made by dispersing micro-containers in the polymeric matrix that release corrosion inhibitors when immersed in water or as a response to environmental pH changes [4,5,6,7]. In the latter case, the repair of damaged areas occurs thanks to the polymerization of the unreacted resin contained in the capsules after their release following the container breaking. The released resin fills up gaps, consolidates, and helps restore scratched areas. Conversely, if the resin itself can flow back into the damaged area and close gaps, the healing mechanism is defined as “intrinsic healing”. In this case, the polymer can reshape itself without additional materials [8,9].

Different from the extrinsic mechanism, the intrinsic one restores the network-mimicking natural systems and allows self-healing to occur multiple times. This strategy is typically based on reversible covalent interactions via disulfide bonds [10], D–A reactions [11,12,13,14], non-covalent interactions via metal-ligands [15], and hydrogen bonds [8,16].

Among intrinsic, self-healing polymers, epoxy resins modified by D–A adducts are promising as protective coatings in aggressive environments. In fact, during mending treatment, D–A epoxies break down into reactive intermediates that are equally stable in aggressive environments. Moreover, even after several damage-healing cycles, these polymers preserve their chemical stability and their 3D network, thus ensuring the coating’s functionality [17,18].

In the specific field of corrosion protection, an additional feature can be integrated by embedding active anti-corrosion inhibitors. The resulting coatings can autonomously protect the metal against corrosion [19].

In general, the most common anti-corrosion inhibitors used are hexavalent chromium [20], benzotriazole (BTA) [21,22], and the more promising CeO_2_ nanoparticles [23], which despite the lower toxicity, do not overcome the risk that the incorporation of nano- and micro-additives may compromise the integrity of the coating.

In this paper, we focused on an epoxy coating with intrinsic mendable ability triggered by moderate external stimuli. The self-healing performance was evaluated through EIS tests and morphological observations. Likewise, a polycaprolactone (PCL)/polypyrrole (PPy)-based coating was found to be able to restore the anticorrosion barrier function when subject to NIR irradiation [23]. In fact, the presence of a photothermal layer of PPy enables the melting of PCL under NIR irradiation. However, the use of a low-melting thermoplastic with reduced resistance to water and solvents greatly limits the applications of the proposed system.

The effectiveness of epoxy coating as a barrier film for corrosion protection for metals is well established [24]. In the present case, water is the transmission medium of oxygen and ions. A methodology to correlate the water uptake and the diffusivity to the coating capacitance was developed by Monetta et al. [25]. Their assumption is that water diffusion in polymers is regarded as an ideal Fick’s process, i.e., water only diffuses in the micro-pores of polymers.

The present work reports the use of a D–A-modified epoxy resin as a protective coating against corrosion for metallic substrates. The capability of the resin to heal small damages has been assessed through vibrational spectroscopy and microscopy. The effect of the thermal treatment on mechanical and morphological performance has been investigated using nanoindentation. The effectiveness of the self-healing polymer to restore protection has been evaluated through electrochemical impedance on the undamaged and scratched/healed samples after exposure to a saline solution. Studies on corrosion resistance are performed in marine environments, given the relevant issue of integrity management of steel structures, such as ships, offshore platforms, bridges, and gas/oil submarine pipelines.

## 2. Materials and Methods

### 2.1. Self-Healing Epoxy Preparation and Coating Application

1,1-methylenedi-4,1phenylene-bismaleimmide, furfuryl glycidyl ether, O,O-bis(2-aminopropyl) polypropylene glycol-block-polyethylene glycol-block-polypropylene glycol (Jeff), 4,4′-diaminodiphenylmethane (DDM), and acetonitrile, hydrochloric acid (HCl), sodium hydroxide (NaOH) were acquired (Sigma Aldrich, St. Louis, MO, USA) and used without further purification; DGEBA EC01 (epoxy equivalent weight ~187 g/eq) was provided by Elantas.

Q235 steel (0.30 wt% in carbon) substrates were chosen for corrosion testing because it is widely used in structural fields. Moreover, its corrosion evaluation is simple, since carbon steel is more prone to corrosion than stainless steel and the oxidation product is a brown substance visible to the naked eye.

The active adduct (2Ph2Epo) and the related self-healing resin (2Ph2Epo65) were synthesized and prepared following a protocol previously developed by the authors [12,26]. The self-healing epoxy formulation—2Ph2Epo and DGEBA (65:35 mol ratio)—was added to a mixture of DDM and Jeff (60:40 mol ratio) as curing agents. The reference sample was prepared using DGEBA and mixed with the same curative (DDM 60: Jeff 40). Both resin formulations were processed, using acetonitrile as the solvent, with a vacuum centrifugal planetary mixer (Thinky Mixer ARV-310) for component dispersion, mixing (2 min at 1500 rpm), and degassing.

The self-healing cycle was constituted of two stages: a first treatment at 120 °C for 20 min—which reduced the crosslink density and promoted molecular recombination—followed by annealing at 90 °C for 6 h—which led to a new network formation [12]. Metal substrates were coated by solvent casting the aforementioned mixture and curing it at 90 °C for 2 h, resulting in 120 μm thick polymer films covered.

Two sets were prepared: the one—with self-healing epoxy resin—to be evaluated, and the other—with DGEBA-based resin—as a reference. The samples, which are called damaged, were scratched with a 3 cm long mark.

### 2.2. Experimental Characterization

Infrared spectroscopy was carried out using a Perkin Elmer Frontier FTIR spectrometer (Waltham, MA, USA) with a single-reflection, universal ATR-IR accessory. The instrument is equipped with a diamond crystal with a 45° incidence to the IR beam. The spectra were collected with a resolution of 4 cm^−1^; a total of 32 scans were averaged for each sample in the range between 1750 and 650 cm^−1^.

Raman spectroscopy was performed using a Renishaw inVia Reflex Raman spectrometer—in backscattering configuration—equipped with a laser source with a 785 nm wavelength (Wotton-under-Edge, UK). Spectra were collected in the range between 1900 and 600 cm^−1^.

To investigate the morphological recovery of coatings, scratches, with an average length of 30 mm, were made on samples using a sharp scalpel. Sample micrographs were recorded by an Olympus BX 51M optical microscope (Tokyo, Japan), equipped with Linkam THM600 hot stage. Images were acquired with 10× magnification, in a bright field, and by recording the morphological changes in the samples by applying a proper thermal stimulus.

Three-point bending mechanical tests were carried out with a DTMA Q800 (TA Instruments (New Castle, DE, USA), with a three-point bending small fixture (span 20 mm), according to ASTM D790 standards, on samples with dimensions of 25 × 8 × 1 mm^3^.

Nanoindentation tests were performed using a NanoTest platform (produced by Micro Materials Ltd., Wrexham, UK), that monitors the dynamic load and displacement of a three-sided, pyramidal, diamond indenter, Berkovich tip with a radius of about 100 nm. The indentation test was performed in force-controlled mode by applying a load in the range of 0.03–1.00 mN with a speed of 1 mN/s. The loading and unloading curves were recorded. To avoid substrate influences on the measurements, the load was set up to avoid an indentation depth higher than 10% of the sample thickness. When the load was removed, the indenter tip was pushed away to recover its initial condition. The reduced modulus (*E_r_*) was proportional to the slope of the unloading curve; the sample stiffness was correlated to the Poisson modulus and the tip shape (Equation (1)) [27]:(1)$$E_{r}=\frac{\sqrt{\pi \;}\;S}{2\;\beta \;\sqrt{A}}=\left[\frac{\left(1-\upsilon_{i}^{2}\right)}{E_{i}}+\frac{\left(1-\upsilon_{S}^{2}\right)}{E_{S}}\right]$$
where *A* is the contact area, β the shape constant (1.034 for a Berkovich indenter), and *S* is the unloading stiffness at maximum load. The elastic modulus *(E)* and Poisson’s ratio (*ν*) are indexed with “i” and “s” subscripts referring to the diamond indenter and the specimen, respectively. *E*_i_ was 1140 GPa, *ν*_i_ was 0.07, and the *ν*_s_ was supposed as 0.35 for the D–A-modified epoxy resin.

Steel substrates were coated with thin epoxy resin film (thickness of 120 µm) through solvent casting. Scratches, with an average length of 30 mm, were made on the coating to expose the metal surface to an aggressive environment.

As a start, the corrosion resistance was morphologically evaluated by subjecting the samples to a saline solution and comparing the healed sample to the pristine and non-treated, damaged ones. A 5 ± 1% by weight aqueous solution of sodium chloride was prepared according to Test Method ASTM B117—16. After setting the temperature at 23 ± 3 °C, the pH of the solution was set between 6.5 and 7.2 using diluted reagent grade HCl and reagent grade NaOH.

EIS experimental tests were carried out at the corrosion potential using a Solartron 1255 frequency response analyzer connected to an EG&G 273 potentiostat. All measurements were performed in a three-electrode system (a saturated calomel electrode as a reference, a platinum electrode as a counter, and the coated substrate as a working electrode). The electrolytic immersion cell, with a circular shape of 3.14 cm^2^ area, was placed on the coated substrate and it was filled with an electrolyte solution (NaCl 3.5%wt) during the experiments.

## 3. Results and Discussion

### 3.1. Diels–Alder Epoxy Resin’s Self-Healing Assessment

A previously developed self-healing epoxy was prepared according to the synthetic route set up by the authors [12]. In brief, the self-healing epoxy resin was formulated as a mixture of 65% mol of thermoreversible polymer and 35% mol of commercial DGEBA (Figure 1) to achieve the thermal and dimensional stability of the system during healing treatments.

The recovery ability of the resin was morphologically evaluated through optical microscopy observation, as a generally accredited technique. Indeed, the capability to self-heal small damages is due to local molecular mobility, triggered by the temperature—higher than *T*g (88 °C)—that promotes the retro Diels–Alder (rD–A) reaction (120 °C) and reduces the crosslink density. In Figure 2a–c, the effects of the thermic treatment are reported. In detail, a superficial scratch 80 μm wide (a) was reduced after heating at 120 °C for 5 min (b) and disappeared after 20 min, resulting in a complete recovery (c) if damage did not cause the removal of the material for the entire sample thickness.

It is noteworthy that the entire healing procedure consisted of first heating at 120 °C for 20 min—to promote molecular mobility—followed by an annealing treatment at 90 °C for 6 h to reconstitute a 3D crosslinked network [17].

Spectroscopic analyses—FT-IR and Raman—were performed to assess the resin’s mendable ability after the complete self-healing process. The failure event was simulated by heating the compound at 120 °C for 20 min, opening all reversible bonds.

Figure 2d shows the FT-IR spectra of the three different rearrangements, in which the D–A crosslinked resin is reorganized by specific thermal treatments: (a) virgin resin, (b) after heating at 120 °C for 20 min and quenching at room temperature, (c) after heating at 120 °C and further annealing at 90 °C for 6 h.

The peak at 1700 cm^−1^, peculiar of C=O stretching of maleimide, was selected as an invariant reference peak. The bands at 688 cm^−1^—related to C=C stretching vibration in the maleimide ring—and at 1146 cm^−1^—attributed to the C–H bond linked to C=C—were susceptible to change with the progress of the D–A and rD–A reactions. In detail, the 688 cm^−1^ peak was not observed in the just crosslinked resin (a); it was detected in the b-curve—after thermal treatment at 120 °C—and it was, again, negligible after the complete healing cycle (c). Similarly, the peak at 1146 cm^−1^, not present in the a-curve, was recorded in the sample heated at 120 °C (b) and disappeared after the annealing (c). Thus, the treatment at 120 °C—favoring the rD–A reaction—induced the D–A adduct rupture and the formation of maleimide and furan-derivatives. Thus, as expected, the absorption intensity of the characteristic peaks of D–A products increased. Further annealing at 90 °C promotes the D–A reaction, which generates D–A adducts. Maleimide and furan groups were consumed during the reaction, so the intensity of their specific peaks was reduced. As a result, the FT-IR spectrum, after the complete self-healing cycle, was very similar to that of virgin resin.

Likewise, Raman analysis was performed to monitor the self-healing process through defined temperature treatments. The peak at 1612 cm^−1^, which is related to C-C aromatic ring chain vibration, was chosen as the invariant reference in the investigation. Instead, the peak at 1501 cm^−1^, attributable to the C=C stretching vibration of the furan ring [28,29], was used to evaluate the reactions’ progress in D–A resin. Raman spectra, shown in Figure 2e, report the curves of self-healing resin in the three characteristic conditions: (a) as crosslinked, (b) after thermal treatment at 120 °C for 20 min and quenching at room temperature, and (c) after additional annealing at 90 °C.

The peak at 1501 cm^−1^ increased in intensity after the first thermal treatment at 120 °C (b-curve), due to the formation of a furan derivative as a product of the rD–A reaction, while this band was reduced after subsequent annealing at 90 °C (c-curve) because of the D–A recombination between furan derivative and maleimide. Therefore, the very similar peak intensity at 1500 cm^−1^ between the a and c spectra can be considered network restoration evidence.

The multiple healing features of the resin were assessed by mechanical tests carried out on the pristine and healed resin. After two complete failure-healing cycles, 2Ph2Epo65 still recovered from damages, exhibiting the virgin stiffness (2.75 GPa), while the strength of the resin decreased as the healing cycles progressed, probably due to residual micro-defects acting as crack initiators in subsequent tests.

Moreover, the evaluation of the resin’s mending capacity was performed using the nanoindentation platform. The sample was tested in the characteristic conditions: virgin; after heating at 120 °C, and finally, healed (annealing at 90 °C). For each test, ten indentation loads, from 50 mN to 250 mN, were performed. In Figure 2g, reporting the final load cycle for each of the three conditions, the elastic modulus was proportional to the slope of the unloading curve; the hardness was measured as the ratio of the maximum load exerted to the effective contact area [27]. The elastic modulus was calculated by fitting the unloading curve with a power-law fit to estimate the contact compliance, C_s_, and the contact depth, *h_c_*, i.e., the difference between the indentation at maximum load and the depth at a zero load. Table 1 reports the average values of the reduced modulus and hardness recorded for each material condition in nanoindentation tests. The system showed good recovery efficiency: the stiffness recorded for the healed material was 83.8% of the virgin one; even the hardness was completely recovered (0.21 ± 0.06 GPa) by heat treatments. It is worth remarking that the elastic modulus calculated from reduced modulus, measuring the depth by indentation, could invariantly be 5–20% higher than that of the tensile, as highlighted by Zheng et al. [30].

### 3.2. Corrosion Resistance Evaluation

The efficiency of the sample to recover protective functionality against the harsh environment through self-healing was evaluated in comparison with the performance of a conventional epoxy resin and a damaged specimen without any remedial treatment.

It is well known that conventional epoxy coatings lose their barrier efficiency in the presence of damage that exposes the metal substrate, where corrosion is prone. A steel substrate protected with intact commercial epoxy coating resists corrosion for a long time when immersed in a saline solution (Figure 3a). If, on the other hand, the coating is scratched, or otherwise damaged, the metal will oxidize rapidly after exposure to a saline solution for a few hours (Figure 3b). Blisters and delamination occurred in 4 days (Figure 3c). Similarly, a deeply cut 2Ph2Epo65 coated sample (Figure 3d) exhibited oxidized spots after being exposed to the saline solution for 3 h (Figure 3e), and after 92 h, additional blistering and delamination were observed on the edges of the scratches (Figure 3f). Further, a damaged 2P2Epo65-coated sample (Figure 3g) subsequently recovered using a healing treatment (20 min @ 120 °C and 6 h @ 90 °C), the healed sample offered the same protection level as the unscratched pristine coating. The recovered sample showed no corrosion damage (Figure 3h) even after the maximum examination time (92 h), as shown in Figure 3i.

Although repair of morphological damage can restore the protective barrier function, further annealing is required to restore crosslink density, which affects the mechanical stability and enhances the durability of the barrier film. Therefore, the morphological observation of scratch damages is not adequate for evaluating the effectiveness of corrosion protection, while water content absorption analysis can pursue quantitative information about the performance of the film during immersion in a saline solution. The electrochemical impedance method is an effective analytical methodology for the assessment of coating performance. It has been widely used to measure organic coatings’ corrosion protection capacity on metallic surfaces [31]. In the case of an insulating polymeric film on a conductive surface, EIS results have been analyzed by considering a two-time constant electrical equivalent circuit [32]. Each parallel in this model (inset in Figure 4b) describes an interface (polymer/solution and metal/solution).

Through the empirical Brasher–Kingsbury equation (Equation (2)) [33,34], it is possible to estimate the water uptake:(2)ϕ=logCtC0logεH2O
where *C_t_* is the coating capacitance at time *t*, *C*_0_ is the capacitance of the “dry” coating and ε*_H_*_2_*_O_* is the water dielectric constant. Figure 4 compares the pristine (scratch-free) sample with one scratched and thermally treated (20 min at 120 °C and 6 h at 90 °C) to activate the healing mechanism. Both the systems showed the same trend, with the phase angles being very close to 90 °C and a flattening of the water absorption curve after 25 h of exposure. In addition, the two samples exhibited similar slopes in the initial stage. This confirms that the healing treatment has fixed the dented area (without altering the water absorption rates and time to saturation), potentially restoring the coating protection. In detail, the coefficient of absorption and water diffusion in the polymer coating can be estimated by analyzing the capacitance evolution. At the initial exposure, the water diffusion is homogeneous within the coating: the graph shows a linear trend according to the Fickian absorption model, where the slope of the capacitance curve (*C* vs. √*t*, Figure 4) is proportional to the diffusion coefficient D (Equation (3)) [35,36]:(3)$$D=\frac{\pi \;\Psi^{2}L^{2}}{16}$$
where Ψ is the capacitance slope and *L* is the coating thickness.

The diffusion observed after healing (1.6 × 10^−6^ cm^2^/s) is close to the pristine diffusion coefficient (3.1 × 10^−6^ cm^2^/s).

A deflection from the linear behavior was detected approaching the saturation at 25 h (300 s^½^). The data depicted in Figure 4 show that the healing treatment induced not only the morphological restoration of the coating by filling the gap left by the scratch, but also recovered the functionality of the protective coating that returns to its pristine diffusive behavior.

## 4. Conclusions

In conclusion, we have proposed a self-mendable epoxy coating to protect metal substrates from environmental exposure. Here, the self-healing ability of the 2Ph2Epo65 formulation has been evaluated. The spectroscopic, mechanical, and nanomechanical analyses, performed on virgin and healed samples, showed satisfactory morphological and mechanical recovery. The FT-IR and Raman analyses confirmed self-healing ability through the recombination of D–A adducts and subsequent restoration of the crosslinked network. The mechanical tests reported pristine rigidity (2.75 GPa) after two complete healing cycles post-fracture.

Therefore, the effectiveness of self-healing epoxy resin as a corrosion barrier has been assessed. The characteristics of the coating film have been evaluated through optical microscopy and EIS investigations after recovering from serious damage and additional exposure to the corrosive environment. The complete self-healing procedure (heating at 120 °C, and subsequent treatment at 90 °C) allowed for the recovery of the protective efficiency of the coating against corrosion in the saline solution. These results confirm the suitability of coatings with self-healing properties as adhesives and corrosion-resistant functional coatings.

## Figures and Tables

**Figure 1 materials-16-01775-f001:**
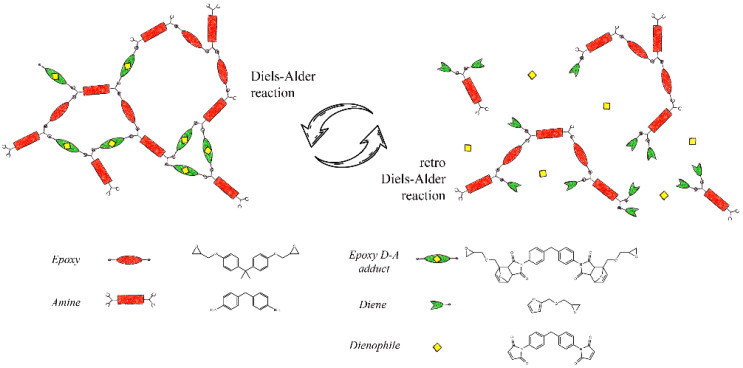
Thermoreversible network induced by D–A adducts.

**Figure 2 materials-16-01775-f002:**
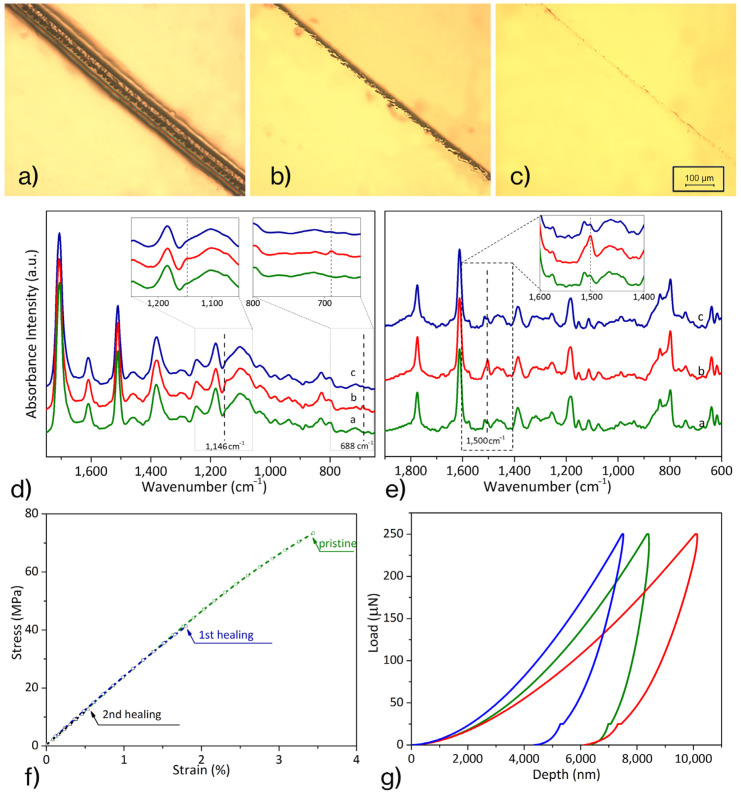
Scratch recovery by the thermal treatment: (**a**) scratched sample, (**b**) after 20 min at 120 °C, (**c**) after a further 6 h at 90 °C; spectra, FT-IR (**d**) and RAMAN (**e**), of 2Ph2Epo65 in the three diverse conditions: as prepared (green, a-curve), after heating at 120 °C for 20 min (red, b-curve), after further treatment at 90 °C for 6 h (blue, c-curve); (**f**) mechanical performances after multiple healing; (**g**) evolution of mechanical properties (nanoindentation) as the effect of thermal treatments: as prepared (green), post-heating at 120 °C (red), and further treatment at 90 °C (blue).

**Figure 3 materials-16-01775-f003:**
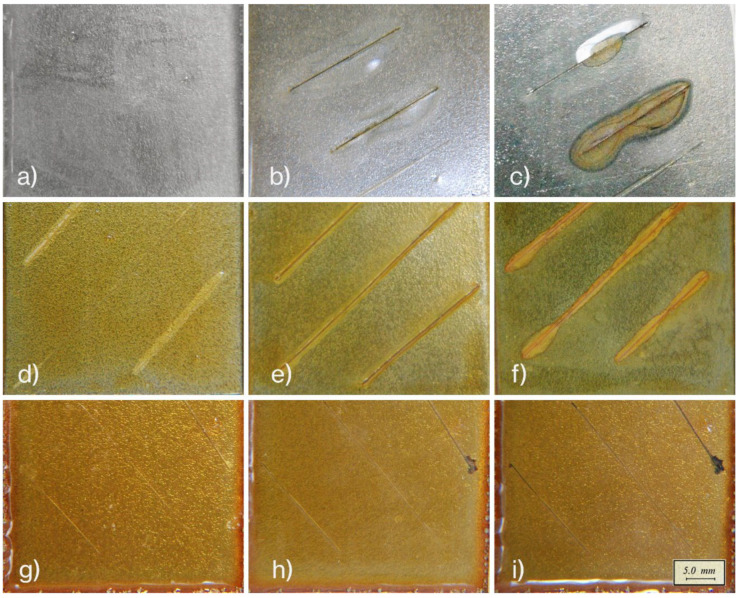
Corrosion resistance of coatings in a saline solution: conventional epoxy coating (**a**), scratched and after subsequent exposure to the saline solution for 3 h (**b**) and 92 h (**c**); untreated self-healing epoxy resin just damaged (**d**), after exposure to the saline solution for 3 h (**e**) and 92 h (**f**); recovered self-healing epoxy resin just immersed (**g**), after exposure to the saline solution for 3 h (**h**) and 92 h (**i**).

**Figure 4 materials-16-01775-f004:**
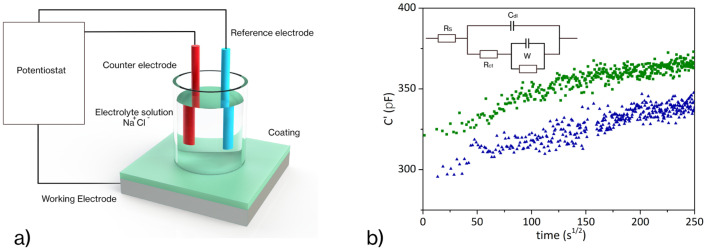
EIS experiment set-up (**a**); water absorption curves (**b**) of the coating as a function of time for pristine coating (green squares) and after healing treatment (blue triangles); with the Randles equivalent circuit in the inset: resistance of solution (Rs), double layer capacitance at the surface of the electrode (CdI), charge transfer resistance (Rct), and Warburg element (W).

**Table 1 materials-16-01775-t001:** Nanoindentation tests on 2Ph2Epo65 resin system.

Description	Elastic Modulus, GPa	Hardness, GPa
Pristine 2Ph2Epo65	4.89 ± 0.35	0.19 ± 0.02
After 20 min @ 120 °C	2.60 ± 0.29	0.16 ± 0.02
Additional 6 h @ 90 °C	4.10 ± 0.63	0.21 ± 0.06

## Data Availability

Not available.

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
