# Peer review of "Thermally Mendable Self-Healing Epoxy Coating for Corrosion Protection in Marine Environments"

_materials, 2023, doi:10.3390/ma16051775_

Round 1

Reviewer 1 Report

Reviewer # :  The authors reported the interesting results and conducted a significant work. The manuscript was well written and organized. However, there existed several issues that should be revised.

1.     In the title specify metal and corrosive medium

2.     Reformulate the abstract in order to clearly show the strengths of this work.

3.     How does this work differ from the previously reported several classes of coating based the epoxy ??? Novelty of the study should be further highlighted.

4.     The choice of substrate and corrosive medium must be justified.

5.     The experimental part must be detailed.

6.     Why you did not use other more powerful techniques in characterization?

7.     Why you chose the 92h exposure time and also chose the temperature.

8.     Comparison with previous works are not reported.

Thus, the manuscript should experience the major revision before acceptance.

Reviewer 2 Report

Re.:                materials-2185355-peer-review-v1.pdf

Title:               Thermally mendable self-healing epoxy for corrosion protective coatings

Author:           Eugenio Amendola, Barbara Palmieri, Stefania Dello Iacono and Alfonso Martone

General Statement

The paper deals with self-repairing protective coatings, a topic that is currently one of the mainstreams of research in materials engineering and polymer chemistry. In addition to its timeliness, it is also characterized by the originality of the data presented - although they deal with a special case of a material structure developed by the authors and are causal in nature. In my opinion the paper is well organized and written with very few minor editing errors. My only reservation is that the paper does not specify how the authors had determined parameters (temperature and time for subsequent steps) of the recovery process. Are these parameters had been optimized in some way?  I believe that the authors' reference to this problem would increase the value of the work. Even in the absence of supplementing the work with appropriate commentary, I recommend it for publication.

Reviewer 3 Report

Comments to authors

1-      The preparation of self healing coating in the experimental part is totally unclear and need more explanation for repeatability.

2-      The reference epoxy polymer is not identified, source and structure.

3-      Figure 4 is so confusing, the authors mentioned that they used three electrode system for EIS meausrment which is not obvious in this figure.

4-      The fitting circuit is need to be identified on the figure with each component.

5-      Authors encouraged to add nyquist plot for more explanation about corrosion protection behavior.

Round 2
